# Semi-Synthesis of C-Ring Cyclopropyl Analogues of Fraxinellone and Their Insecticidal Activity Against *Mythimna separata* Walker

**DOI:** 10.3390/molecules25051109

**Published:** 2020-03-02

**Authors:** Xiao-Jun Yang, Qing-Miao Dong, Min-Ran Wang, Jiang-Jiang Tang

**Affiliations:** 1School of Chemistry & Chemical Engineering, Yanan University, Yanan 716000, China; yangxiaojun2002@126.com; 2Shaanxi Key Laboratory of Natural Products & Chemical Biology, College of Chemistry & Pharmacy, Northwest A&F University, Yangling 712100, China; qingmiaodong@nwafu.edu.cn (Q.-M.D.); 13833683803@163.com (M.-R.W.)

**Keywords:** fraxinellone, insecticidal agent, cyclopropanation

## Abstract

Fraxinellone (**1**) is a naturally occurring degraded limonoid isolated from Meliaceae and Rutaceae plants. As a potential natural-product-based insecticidal agent, fraxinellone has been structurally modified to improve its activity. Furan ring of fraxinellone is critical in exhibiting its insecticidal activity, but with few modifications. Herein, C-ring-modified cyclopropyl analogues were semi-synthesized by Rh(II)-catalyzed cyclopropanation. The structures of the target compounds were well characterized by NMR and HRMS. The precise three-dimensional structural information of **3a** was established by X-ray crystallography. Their insecticidal activity was evaluated against *Mythimna separata* Walker by a leaf-dipping method. Compound **3c** exhibited stronger insecticidal activity than **1** and toosendanin against *M. separata* with teratogenic symptoms during the different periods, implying that cyclopropanation of the furan ring could strengthen the insecticidal activity of fraxinellone.

## 1. Introduction

Natural products (NPs) continue to provide a rich source for drug discovery [1,2,3,4,5,6,7,8]. In the past decades of controlling agricultural pests, NPs also played a key role in the discovery of novel pesticides owing to their potential target sites, low toxicity, and environment-friendly characteristics [4,5,6,7,8]. In order to improve the hit rate, NPs were always synthesized with diverse structures in a screening molecules library [9].

Fraxinellone (**1**, Figure 1) containing three rings (labeled A–C) is a degraded limonoid isolated from barks and roots of *Dictamnus dasycarpus* Turcz. and exhibits interesting insecticidal activity [10,11]. The activity relationships of fraxinellone showed that the furan ring (C-ring) is essential for its insecticidal activities [12]. Several NPs examples with the furan ring as an important pharmacophore exhibit various bioactivities [4,13,14,15,16,17]. For example, toosendanin (Figure 1) showed potent antifeedant and growth inhibitory effects against armyworm *Mythimna separata* and cutworm *Peridroma saucia*, in which the furan ring is critical [16,17].

To improve the agrochemical activities of fraxinellone, structural modifications of its A-ring and B-ring have been carried out by Xu group [18], and some fraxinellone-based esters and hydrazones analogues at C-4/C-10 position (A-ring) displayed stronger insecticidal activities against *Mythimna separata* Walker and *Plutella xylostella* Linnaeus. However, the furan ring of fraxinellone remains with few modifications. Recently, the modification of aromatic (furan ring) functionalities was reported by us [19,20]. A series of C-ring selective brominations and further palladium-catalyzed transformations analogues of fraxinellone have been prepared. To our delight, some compounds displayed more potent insecticidal activity than toosendanin. On the basis of the above furan-modified results and in our endeavor aiming at finding more active natural-product-based insecticidal hits [19,21,22,23,24], herein, we semi-synthesized C-ring cyclopropyl analogues of fraxinellone by Rh(II)-catalyzed cyclopropanation as insecticidal agents against *M. separata.*

## 2. Results and Discussion

### 2.1. Semi-Synthesis

An attractive approach to utilize the furan ring for the generation of resourceful analogues is the [2 + 1] addition of carbenes [25,26]. As shown in Scheme 1, the commercial substrate aromatic ethyl esters firstly reacted with *p*-toluenesulfonyl azide (TsN_3_) to form α-carbonyldiazoesters (**2a**–**2c**) in the presence of base 1,8-diazabicyclo[5.4.0]undec-7-ene (DBU) in high yields (67–90%). Yet, when trimethylamine (Et_3_N) was used as the base, this reaction did not occur. If methyl 2-thiophene acetate was used to react with TsN_3_ under the above reaction conditions, the target compound cannot be prepared.

With acceptor diazoesters (**2a**–**2c**) in hand, fraxinellone (**1**) as a starting brick was converted into the corresponding monocyclopropanated adducts (**3a**–**3c**). Rh_2_(OAc)_4_-catalyzed cyclopropanation of **1** mainly provides stereoselective adducts by thin-layer chromatography (TLC) isolation. The absolute configuration of **3a** was unambiguously established by X-ray crystallography (Figure 2), indicating the cyclopropyl ring of 4’ and 5’ positions was formed in β configuration of the furan ring. Formation of this product can be rationalized on steric ground in the asynchronous concerted cyclopropanation that the reaction of the Rh-bound carbene occurs from the *si* face of the carbene [27,28]. Using the same orientation of attack, the observed stereochemistry is consistent with the attack occurring at the *β*-position for **3b** and **3c** in NOE spectra. Furthermore, the configuration of **3a** indicated the top chiral carbon of the cyclopropyl ring oriented the ester group onto the convex face of the bicycle, which is same as synthetic methodology [29]. Unfortunately, the yields were too low (10–37%), with just 50–60% conversion of diazoesters and about 70% conversion of fraxinellone, perhaps because of the complexity of the fraxinellone structure. The structures of all analogues were determined by NMR and HRMS and were stable for at least five days in solid or in acetone solution at room temperature. Their spectra can be found in the Appendix A.

### 2.2. Insecticidal Activity

The insecticidal activity of all cyclopropanated analogues was tested against *M. separata* Walker by the leaf-dipping method as the mortality rates at 1 mg/mL [30]. Toosendanin, a commercial natural-product-based insecticide, was used as a positive control and leaves treated with acetone alone were used as a blank group. The corrected mortality rate at different stages was outlined in Table 1. Fraxinellone and its prepared analogues exhibited the delayed insecticidal activity against *M. separata*. For example, the mortality rates of **3a** against *M. separata* after 10 and 20 days were 8.3% and 12.5%, respectively. However, it was increased to 29.2% after 35 days with over three-fold of that after 10 days. Among all tested analogues, **3c** displayed stronger insecticidal activity than their precursor **1** and toosendanin. The final mortality rates of **3c** were 34.5%, whereas the final mortality rates of **1** and toosendanin were 17.4% and 33.3%, respectively.

Notably, these analogues have good teratogenic activity on three different stages. During the larval period, the symptoms of the larvae of *M. separata* treated by these compounds were slim and wrinkled bodies (not shown). Many larvae molted to malformed pupae or died in the treated groups during the stage of pupation (Figure 3a). Many malformed moths of the treated groups appeared with imperfect wings during the emergence period (Figure 3b), implying that these analogues might have affected the insect molting hormone [31,32]. These symptoms in the three different periods tested by compounds were consistent with those of fraxinellone funan ring-coupling analogues [19]. This demonstrated that inserting the clopropyl group on C-ring of fraxinellone resulted in more promising analogues.

## 3. Materials and Methods 

### 3.1. General 

All NMR spectra were recorded on a Bruker Advance III 500 instrument (Bruker Daltonics Inc., Bremen, Germany) in CDCl_3_ or CD_3_OD with TMS as internal standard for protons and solvent signals as internal standard for carbon spectra. Chemical shift values are mentioned in δ (ppm) and coupling constants (*J*) are given in Hz. HR-ESI-MS spectra were recorded on an AB Sciex 5600 Triple TOF mass spectrometer (AB SCIEX Inc., Singapore). Melting points (m.p.) were determined on an auto-melting point apparatus (Hanon Instruments Co., Ltd., Jinan, China). Column chromatography (CC) was performed over silica gel (200–300 mesh, Qingdao Marine Chemical Ltd.). All reactions were monitored by thin-layer chromatography (TLC) carried out on pre-coated silica gel GF_254_ plates with a thickness of 0.25 mm (Qingdao Marine Chemical Group, Co.) with UV light (254 nm and 365 nm). All commercial solvents and reagents were freshly purified and dried by standard techniques prior to use. Fraxinellone (**1**) was isolated from root bark of *Dictamnus*
*dasycarpus* and established on the basis of extensive spectroscopic analyses, referring to a previous report [19].

### 3.2. Synthesis of α-Diazocarbonyl Esters (**2a**–**2c**)

The stirring solution of 1.58 mmol phenylacetate in 20 mL acetonitrile (MeCN) was added dropwise 379.1 mg (1.90 mmol) of *p*-toluenesulfonyl azide (TsN_3_) in 4 mL MeCN at room temperature. Then, 337.48 mg of 1,8-Diazabicyclo[5.4.0]undec-7-ene (DBU) (1.58 mmol) was added and the solution was stirred for 15 h. After monitoring the completion of the reaction by TLC, water was added to terminate the reaction. The reaction solution was extracted with anhydrous ether (2 × 30 mL) and the organic phase was dried in anhydrous sodium sulfate (Na_2_SO_4_). The organic phase was concentrated in vacuum to give a yellow oil. A silica gel column (300–400 mesh, petroleum ether: ethyl acetate, 40:1–20:1, *v/v*) was used for column chromatography to obtain the products **2a**–**2c**.

*Methyl 2-diazo-2-phenylacetate* (**2a**): yellow solid, 67% yield. NMR data were consistent with a previous report [29]. 

*Ethyl 2-diazo-2-phenylacetate* (**2b**): yellow solid, 89% yield. ^1^H-NMR (500 MHz, MeOD) δ 7.54 – 7.45 (m, 2H), 7.39 (ddd, *J* = 8.6, 7.4, 1.6 Hz, 2H), 7.19 (td, *J* = 7.4, 1.3 Hz, 1H), 4.32 (qd, *J* = 7.1, 1.8 Hz, 2H), 1.34 (td, *J* = 7.1, 1.5 Hz, 3H); ^13^C-NMR (126 MHz, MeOD) δ 173.49, 166.59, 130.25, 129.90, 129.47 (2C), 126.87, 125.11, 62.13, 49.51, 49.34, 49.17, 49.00, 48.83, 48.66, 48.49, 14.76.

*Ethyl 2-(4-bromophenyl)-2-diazoacetate* (**2c**): yellow solid, 88% yield. ^1^H-NMR (500 MHz, CDCl_3_) δ 7.52 (dd, *J* = 8.9, 2.4 Hz, 1H), 7.39 (dd, *J* = 8.9, 2.4 Hz, 1H), 4.37 (qd, *J* = 7.1, 1.8 Hz, 1H), 1.38 (td, *J* = 7.1, 1.6 Hz, 2H); ^13^C-NMR (126 MHz, CDCl_3_) δ 171.35, 165.10, 132.31 (2C), 125.65 (2C), 125.23, 119.58, 77.67, 77.42, 77.16, 61.50, 14.82.

### 3.3. Synthesis of C-Ring Cyclopropyl Analogues (**3a**–**3c**) of Fraxinellone

First, 40 mg (172.21 μmol) of fraxinellone (**1**) and 0.76 mg (1.72 μmol) of Rh_2_(OAc)_4_ were added into a dry reaction tube sealed with a rubber tap. In the tube, air was replaced four times with argon, and the tube was placed in an ice bath. Then, 378.21 μmol phenyldiazoesters (**2a**–**2c**) dissolved in 4 mL of toluene was added dropwise to the tube and stirred in an ice bath for 13 h. After the reaction was monitored for completion by TLC, the solvent was removed directly and the sample was separated in silica gel column chromatography (300–400 mesh, petroleum ether: ethyl acetate, 40:1–10:1, *v/v*) to afford the analogues **3a**–**3c**.

*Methyl-4-((1R,7aR)-4,7a-dimethyl-3-oxo-1,3,5,6,7,7a-hexahydroisobenzofuran-1-yl)-6-phenyl-2-oxabicyclo[3.1.0]hex-3-ene-6-carboxylate* (**3a**): white solid, 20% yield, R_f_ = 0.25 (petroleum ether: EtOAc = 40:1), m.p. = 141–143 °C. ^1^H-NMR (500 MHz, CDCl_3_) δ 7.33–7.29 (m, 3H, H-2″, H-4″, and H-6″), 7.21 (dd, J = 7.4, 1.9 Hz, 2H, H-3″, and H-5″), 6.05 (d, J = 0.8 Hz, 1H, H-2′), 5.19 (d, J = 6.2 Hz, 1H, H-5′), 4.53 (d, J = 1.7 Hz, 1H, H-8), 3.67 (s, 3H, -Me), 3.18 (d, J = 5.7 Hz, 1H, H-4′), 2.32 (dd, J = 19.7, 6.4 Hz, 1H, H-4), 2.22 (dd, J = 10.9, 7.3 Hz, 1H, H-4), 2.14 (s, 3H, H-10), 2.04 (dt, J = 12.2, 3.3 Hz, 1H, H-5), 1.81-1.93 (m, 2H, H-5, 6), 1.61 (td, J = 12.9, 3.8 Hz, 1H, H-6), 1.02 (s, 3H, H-11); ^13^C-NMR (126 MHz, CDCl_3_) δ173.99, 169.48, 149.04, 144.80, 131.94, 130.03, 128.33, 128.03, 127.35, 113.61, 84.11, 71.29, 52.93, 42.72, 39.01, 32.83, 32.18, 28.54, 20.77, 18.69, 18.61; HR-ESI-MS m/z: Found 403.1496 [M+Na]^+ ^(calcd for C_23_H_24_NaO_5_, 403.1521).

*Ethyl-4-((1R,7aR)-4,7a-dimethyl-3-oxo-1,3,5,6,7,7a-hexahydroisobenzofuran-1-yl)-6-phenyl-2- oxabicyclo[3.1.0]hex-3-ene-6-carboxylate* (**3b**): white solid, 37% yield, R_f_ = 0.21 (petroleum ether: EtOAc = 40:1), m.p. = 144–145 °C. ^1^H-NMR (500 MHz, CDCl_3_) δ 7.22–7.15 (m, 3H, H-2″, H-4″, and H-6″), 7.09 (dd, J = 7.5, 1.8 Hz, 2H, H-3″, and H-5″), 5.93 (d, J = 0.8 Hz, 1H, H-2′), 5.07 (d, J = 5.7 Hz, 1H, H-5′), 4.41 (d, J = 1.7 Hz, 1H, H-8), 4.04–3.98 (m, 2H, Et-H ), 3.05 (d, J = 5.7 Hz, 1H, H-4′), 2.24–2.09 (m, 2H, H-4), 2.02 (s, 3H, H-10), 1.92 (dt, J = 12.2, 3.3 Hz, 1H, H-5), 1.79–1.66 (m, 2H, H-5, and H-6), 1.53–1.47 (m, 1H, H-6), 1.07 (t, J = 7.1 Hz, 3H, Et-H), 0.90 (s, 3H, H-11); ^13^C-NMR (126 MHz, CDCl_3_) δ 173.22, 169.26, 148.69, 144.60, 131.74, 130.07, 128.02, 127.67, 127.28, 113.44, 83.97, 70.96, 61.42, 42.54, 38.53, 32.66, 32.00, 28.50, 20.56, 18.49, 18.44, 14.17; HR-ESI-MS m/z: Found 417.1647 [M+Na]^+ ^(calcd for C_24_H_26_NaO_5_, 417.1678).

*Ethyl-6-(4-bromophenyl)-4-((1R,7aR)-4,7a-dimethyl-3-oxo-1,3,5,6,7,7a-hexahydroisobenzofuran-1-yl)-2- oxabicyclo[3.1.0]hex-3-ene-6-carboxylate* (**3c**): white solid, 10% yield, R_f_ = 0.30 (petroleum ether: EtOAc = 30:1), m.p. = 160–162 °C. ^1^H-NMR (500 MHz, CDCl_3_) δ 7.48 (d, J = 7.5 Hz, 2H, H-3″, and H-5″), 7.37 (d, J = 7.9 Hz, 2H, H-2″, and H-6″), 5.81 (s, 1H, H-2′), 5.11 (d, J = 5.5 Hz, 1H, H-5′), 4.33 (s, 1H, H-8), 4.08–4.03 (m, 2H, Et-H), 3.37 (d, J = 5.5 Hz, 1H, H-4′), 2.23 (dd, J = 19.8, 6.4 Hz, 1H, H-4, and H-10), 2.12–2.05 (m, 4H, H-4, and H-10), 1.81–1.61 (m, 3H, H-5, and H-6), 1.30 (dd, J = 14.4, 7.3 Hz, 1H, H-6), 1.13 (t, J = 7.1 Hz, 3H, Et-H), 1.06 (s, 3H, H-11 ); ^13^C-NMR (126 MHz, CDCl_3_) δ 172.75, 170.04, 148.98, 143.71, 132.03, 129.74, 129.37, 127.14, 121.73, 115.17, 83.22,70.64, 61.79, 44.45, 40.19, 32.34, 31.29, 28.08, 20.44, 18.65, 18.17, 14.36; HR-ESI-MS m/z: Found 495.0762 [M+Na]^+ ^(calcd for C_24_H_25_BrNaO_5_, 495.0783).

### 3.4. X-Ray Experimental of **3a**

Single crystals of **3a** (C_23_H_24_O_5_) were obtained by recrystallization in ethanol. A suitable crystal was selected and analyzed on a SuperNova, Dual, Cu at zero, Eos diffractometer (Bruker Daltonics Inc., Bremen, Germany). The crystal was kept at 293(2) K during data collection. Using Olex2 [33], the structure was solved with the Superflip [34] structure solution program using charge flipping and refined with the ShelXL [35] refinement package using least squares minimisation.

### 3.5. The Insecticidal Activity Assay

The insecticidal activity of **3a**–**3c** was tested as the mortality rate using the leaf-dipping method against the pre-third-instar larvae of *M. separata* using the reported procedure [30,36]. Briefly, 24 pre-third-instar larvae (6 larvae per group) were used for each sample. Each treatment was performed four times. Acetone solutions of samples and toosendanin (positive control) were prepared at 1 mg/mL. Fresh wheat leaf discs (1 × 1 cm) were dipped into the corresponding solution for 3 s, then taken out and dried. Several pieces of treated leaf discs were kept in each six-well plate, which was then placed in a conditioned room (25 ± 2 °C, 65–80% relative humidity (RH), 12 h/12 h (light/dark). After two days, untreated fresh leaves were added to the all dish until the adult pupae emergence. The corrected mortality rates of the tested compounds against *M. separata* Walker were calculated in three different periods by the following formula:(1)$$\begin{array}{l} \;\mathrm{corrected}\;\mathrm{mortality}\;\mathrm{rate}\;(\%)=\frac{\;\mathrm{mortality}\;\mathrm{rate}\;\mathrm{of}\;\mathrm{test}\;-\;\mathrm{mortality}\;\mathrm{rate}\;\mathrm{of}\;\mathrm{control}}{100\%-\mathrm{mortality}\;\mathrm{rate}\;\mathrm{of}\;\mathrm{control}}\times \;100\% \end{array}$$

## 4. Conclusions

Natural-product analogues attract extensive attention because of their high hit rate in the process the discovery of new drugs. To enrich the fraxinellone-based structures, we semi-synthesized C-ring-modified cyclopropyl analogues by Rh(II)-catalyzed cyclopropanation for the first time. Notably, this cyclopropanation of fraxinellone provided products with orienting cyclopropyl ring in β configuration and ester group on the convex face of the bicycle, but without altering other functional groups. Although their yield is not high, the reaction applied the synthetic strategy for modification of diverse furan-containing NPs. An evaluation of insecticidal activity showed that **3c** displayed stronger insecticidal activity than parental counterpart and toosendanin. Their corresponding pharmacological data and target of action are undergoing.

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
