# Peer review of "Semi-Synthesis of C-Ring Cyclopropyl Analogues of Fraxinellone and Their Insecticidal Activity Against Mythimna separata Walker"

_molecules, 2020, doi:10.3390/molecules25051109_

Round 1

Reviewer 1 Report

The manuscript #720763 entitled “semisynthesis of C-ring cyclopropyl analogues of fraxinellone and their insecticidal activity against Mythimma separata Walker” by Tang J.J. et al. describes cyclopropyl analogues of fraxinellone and their insecticidal activity against M. separata.

The manuscript should be revised (major revision) before be accepted for publication in Molecules.

Major point are:

  • why the authors calculated the mortality rate in three different periods (larvae pupae and moth) all together?
  • The authors should calculate the insecticidal activity in different stages of the insect i.e. against larvae, pupae and moth separately. Then they should calculate LC50. Mortality rate (%) is the preliminary stage of the study of insecticidal activity of a compound and it is not enough for a full article.
  • What does it mean the last sentence of page 3 “these observation were consistent with Pd-catalyzed coupling analogues of fraxinellone resulting in the teratogenic activity.”?

A Minor point is:

  • references # 1, 5 and 7 are not appropriate, hence delete them.

Author Response

1. Thanks. Indeed, the LC50 is better description for insecticidal activity of the new compounds and should be calculated. Yet, the results of preliminary mortality rate (%) displayed the analogues did not have a distinguishing improvement, just a slight extent, compared to the positive control. Meanwhile, the symptoms in the three different periods tested by the new analogues were similar to those of Fraxinellone and also provided some information for further investigation. Thus, we did not perform the 100 mg-scale synthesis to evaluate the LC50. The evaluation of preliminary mortality rate (%) for new natural products analogues with few amounts is always published in many prime journals. For example:

1) Guo, Y.; Qu, H.; Zhi, X. Y.; Yu, X.; Yang, C.; Xu, H. Semisynthesis and Insecticidal Activity of Some Fraxinellone Derivatives Modified in the B Ring. J. Agric. Food Chem. 2013, 61, 11937-11944.

2) Dong, Q. M.; Dong, S.; Shen, C.; Cao, Q. H.; Song, M. Y.; He, Q. R.; Wang, X. L.; Yang, X. J.; Tang, J. J.; Gao, J. M. Furan-Site Bromination and Transformations of Fraxinellone as Insecticidal Agents Against Mythimna separata Walker. Sci. Rep. 2018, 8, 8372

2. We have revised this sentence instead by “These symptoms in the three different periods tested by compounds were consistent with those of fraxinellone coupling analogues”

3. We have revised references .

Reviewer 2 Report

A study is reported on the synthesis of cyclopropanated analogs of fraxinellone, which are then evaluated on their with respect to their insecticidal activity. The biological results obtained are significant, which gives credit to the overall approach the authors follow here.

As the key reaction, the rhodium catalyzed cyclopropanation of a furan moiety is reported. This process is in general known, however, the compound the authors apply this reaction to has some additional complexity. The authors conclude that the reaction proceeds regio- and diastereoselectivey, being corroborated by one of Xray structure of the products, and overall in line with previous reports on the title reaction. Unfortunately, the yields are very low (3 reactions, 10-37%), which makes it hard to corroborate that the reactions presented are really selective. Also, judged on the SI, the purity of the final compounds is not really clear. Thus, HPLC traces of the crude and purified products would be desirable, moreover, the authors should explain if full conversion of the diazoester was achieved, and what are then the remaining products.

The experimental part must be improved. It is not clear what the yields are based on (diazoester of compound 1), moreover, actual amounts isolated must be stated (please note that a % yield is already processed data). Rf values of the compounds based on the isolation by chromatography and melting points are needed as well. 

Major language improvement is necessary, the manuscript contains numerous stylistic and grammatical errors.

Author Response

1. Thanks for your suggestion. Indeed, the yields were so low (just 10-37% isolated yields) by rhodium catalyzed cyclopropanation of fraxinellone. During the analysis of TLC, there were points of isolated products and other some smaller polar mingled points. The mingled points are hard to analyze and isolate by HPLC. They may be other impurities with different polar from target products perhaps due to the complexity of fraxinellone. And we also found that the conversion of the diazoester in our conditions (equiv. of fraxinellone: diazoester is 1:1) is not high, just 50-60% conversion, which may be the reason of resulting in the low isolated yield. After the cyclopropanated analogues were isolated by CC, we did not carry out the optimization of synthetic conditions.

We revised the description of results. Please see them in the revised manuscript. The purity of the final compounds was >95% by HPLC analysis and HR-MS. Rf values and melting points were measured and added in the text.

2. We invited a native-speaking English friend to improve the language.

Reviewer 3 Report

A paper reports the investigation on some derivatives of fraxinellone, a potential natural-product-based insecticidal agent. The authors modified the structure of fraxinellone introducing cyclopropane ring to its molecule, by the use of Rh2(AcO)4 catalyzed stereoselective cyclopropanation. Structures of three final compounds have been characterized by NMR (1H and 13C NMR) and HRMS. In the case of one compound, the structure was established by X-ray crystallography. Finally, the authors performed preliminary investigations of the insecticidal activity of obtained compounds.

The paper is prepared carefully, the material is presented clearly and briefly, and contains all important information. I have no objections to this work and recommend its publication as it is, without changes.

Author Response

Thanks for your comments.

Round 2

Reviewer 1 Report

The replies of the authors are sufficient, the manuscript has been improved and now warrants publication in Molecules.

Author Response

Thank you for your comments again.

Reviewer 2 Report

The authors report on the modification of Fraxinellone by rhodium-catalyzed cyclopropanation and the evaluation of the insecticidal activity of the derivatives obtained. Overall, I find the concept followed here interesting, and a somewhat improved insecticidal activity was found compared to the natural product. Thus, I am supportive of the publication of this study, but I have a number of concerns which should be addressed:

  1. The authors claim that the reactions if chemo and diastereoselective. In light of the very low yields obtained in the cyclopropanation, such a claim is tricky. Is this claim corroborated by crude NMR spectra (even the purified compounds still show impurities judging the NMR spectra provided in the SI)? Since the yields are so low, what happens with the remaining starting material? No conversion or unidentified products? If the latter is the case, the claim of diastereoselective cyclopropanation should not be made.
  2. Given the fact that the furan moiety appears to be crucial for biological activity following the arguments made in the introduction, it is somewhat surprising that the cyclopropanated adducts presented in this study should have improved activity. Given the long essay times (days), I wonder what is really tested. Cyclopropanated furans (donor-acceptor substitution) of the type reported here are known to rapidly undergo ring-opening with concurrent rearomatization to give 2- or 3-substituted furans, especially in aqueous solution. I wonder if the real compounds tested here were not the cyclopropanated adducts but the ring-opened and rearomatized compounds (this yields similar compounds the authors have evaluated before, cf. Figure 1 "previous work"). It should be possible to easily test this hypothesis by transforming the cyclopropanated adducts to the ring-opened furans.

Author Response

Q1.
Response: Thank you for your comments. Actually, after points of isolated products were isolated by TLC, other some smaller polar mingled points were too hard to isolate by TLC or HPLC. NMR spectra provided in the SI showed few impurities should not be cyclopropanated analogues according to ppm judgments, and the few impurities may be from solvents. Of course, the purity of the final compounds was ok with >95%. Based on our original experimental records, conversion of starting material is about 70% and conversion of diazoesters is just 50-60%. In this work, we focused more on the C-ring transformed products of Fraxinellone and not on yield, so we did not carry out the conditions optimization. In the revised manuscript, in order to avoid providing uncertain information on the diastereoselective, we revised the description of stereoselective cyclopropanation and cancelled the definitive words of diastereoselective.

Q2.
Response:  This is a good suggestion for our further investigation. In process of insecticidal activity assay, samples and toosendanin (positive control) were prepared in acetone solutions. Fresh wheat leaf discs (1×1 cm) were dipped into the solution for 3 s, then taken out and dried. After treatment of 2 days, untreated fresh leaves were added to the all dish until the adult pupae emergence. In the acetone solutions or solids within 2 days, no change of cyclopropanated structures occurred by ring-opening or others. Due to few amounts of natural products analogues, we will evaluate the activity of ring-opening analogues after the new isolation with > 1 g amounts of fraxinellone and further modification. Thanks for your idea again.